# Association of Non-Alcoholic Fatty Liver Disease and Metabolic-Associated Fatty Liver Disease with COVID-19-Related Intensive Care Unit Outcomes: A Systematic Review and Meta-Analysis

**DOI:** 10.3390/medicina59071239

**Published:** 2023-07-03

**Authors:** Gowthami Sai Kogilathota Jagirdhar, Harsha Pattnaik, Akshat Banga, Rakhtan K. Qasba, Kaanthi Rama, Shiva Teja Reddy, Anna Carolina Flumignan Bucharles, Rahul Kashyap, Praveen Reddy Elmati, Vikas Bansal, Yatinder Bains, Theodore DaCosta, Salim Surani

**Affiliations:** 1Department of Medicine, Saint Michaels Medical Center, Newark, NJ 07102, USA; 2Lady Hardinge Medical College, New Delhi 110001, India; 3Sawai Man Singh Medical College, Jaipur 302004, India; 4Green Life Medical College and Hospital, Dhaka 1205, Bangladesh; 5Gandhi Medical College, Telangana 500025, India; 6Universidade Positivo, Curitiba 81280-330, PR, Brazil; 7Critical Care Medicine, Department of Anesthesiology, Mayo Clinic, Rochester, MN 55905, USA; 8Interventional Pain Medicine, University of Louisville, Louisville, KY 40202, USA; 9Division of Nephrology and Critical Care Medicine, Department of Internal Medicine, Mayo Clinic, Rochester, MN 55905, USA; 10Department of Gastroenterology, Saint Michaels Medical Center, Newark, NJ 07102, USA; 11Pulmonary, Critical Care & Pharmacy, Texas A&M University, College Station, TX 79016, USA

**Keywords:** non-alcoholic fatty liver, fatty liver, coronavirus, COVID-19, metabolic-associated fatty liver, critical care, intensive care unit, NAFLD, MAFLD

## Abstract

*Background and Objective*: The association of non-alcoholic fatty liver disease (NAFLD) and metabolic-associated fatty liver disease (MAFLD) with intensive care unit (ICU) admissions and the need for mechanical ventilation and disease severity in COVID-19 patients. *Material and Methods*: A systematic literature review was conducted on the databases: Cochrane, Embase, PubMed, ScienceDirect, and the Web of Science from January 2019 to June 2022. Studies evaluating MAFLD using laboratory methods, non-invasive imaging, or liver biopsy were included. The study protocol was registered in PROSPERO (ID CRD42022313259), and PRISMA guidelines were followed. The NIH quality assessment tool was used for quality assessment. RevMan version 5.3 software was used for pooled analysis. A sensitivity analysis was performed to assess the result’s stability. *Results*: A total of 37,974 patients from 17 studies were assessed for the association between MAFLD and ICU admission. A total of 3396 COVID-19 patients required ICU admission: 1236 (20.41%) in the MAFLD group and 2160 (6.77%) in the non-MAFLD group. The odds ratio was 1.86 for ICU admission, *p* = 0.007, and a (95% CI) of [1.18–2.91]. A total of 37,166 patients from 13 studies were included in the need for invasive mechanical ventilation analysis. A total of 1676 patients required mechanical ventilation: 805 in the MAFLD group (14.20% of all MAFLD patients) and 871 patients in the non-MAFLD group (2.76% of all non-MAFLD patients). The odds ratio was 2.05, *p* = 0.02, and a (95% CI) of [1.12–3.74]. A total of 5286 patients from 14 studies were included in the COVID-19 disease severity analysis. Severe COVID-19 was seen in 1623 patients, with 33.17% (901/2716) of MAFLD patients and 28.09% (722/2570) of non-MAFLD patients having severe disease. The odds ratio was 1.59 for disease severity, *p* = 0.010, and a (95% CI) of [1.12–2.26]. *Conclusions*: Our meta-analysis suggests that there are significantly increased odds of ICU admissions, a need for invasive mechanical ventilation, and disease severity in MAFLD patients who acquire COVID-19.

## 1. Introduction

Since its inception back in January of 2021, COVID-19 has proved to be a worldwide pandemic, amassing a total of six and a half million deaths as of September 2022 and more than 600 million confirmed cases worldwide [1]. Most patients present with milder forms of this disease and a wide range of signs and symptoms, mostly self-limiting. Even though COVID-19 is mainly associated with pulmonary signs and symptoms, recent reports suggest otherwise. Extrapulmonary manifestations, which are hepatological [2], neurological [3], gastrointestinal [4], and cardiac [5], also seem to be associated with severity. Recent studies show COVID-19 severity is associated with chronic health conditions [6] where conditions such as diabetes, hypertension, chronic obstructive pulmonary disease, and obesity were identified as clinical risk factors. Interestingly, a recent cross-sectional study from Germany suggested that obesity/visceral adiposity and upper abdominal circumference increased the risk of ICU admissions in COVID-19 patients [7]. Moreover, patients identified under the umbrella term of metabolic syndrome (MetS) were reported on many occasions to be associated with severe outcomes in COVID-19 [8,9].

Metabolic-associated fatty liver disease (MAFLD) and non-alcoholic fatty liver disease (NAFLD) are alike, according to a recent consensus of experts [2,10,11]. They are now considered to be the hepatic manifestations of MetS [12]. NAFLD is prevalent in one in every fourth individual worldwide and is therefore considered to be one of the most common causes of chronic liver disease (CLD) [13]. Studies have linked NAFLD to severe COVID-19 [14,15]. NAFLD patients have a higher chance of liver function being abnormal and a higher risk of COVID-19 disease progression compared to patients without NAFLD [16]. The risk of having severe COVID-19 in patients with MAFLD was more than two-fold higher than without MAFLD for patients under 60 years of age [17]. Adding to that, on many different occasions, patients with MetS and abnormal liver function were linked with increased ICU admission and a severe COVID-19 disease course [9,18].

Therefore, the literature shows that NAFLD and MAFLD could have an increased risk of severe COVID-19, requiring ICU-level supervision and care. However, there is also conflicting data on this relationship. Therefore, our systematic review and meta-analysis aimed to assess the association of NAFLD and MAFLD with ICU admission, mechanical ventilation, and the severity of COVID-19 in patients by meta-analyzing the data in the available literature.

## 2. Methods and Materials

This systematic review and meta-analysis was performed following the PRISMA (Preferred Reporting Items for Systematic Reviews and Meta-Analyses) statement [19], as indicated in the PRISMA checklist, and was registered with PROSPERO (ID CRD42022313259; www.crd.york.ac.uk/prospero accessed on 10 June 2023).

### 2.1. Search and Selection

A systematic search was performed using a search strategy in six different databases, including PubMed, Cochrane, Embase, Science Direct, and the Web of Science, with a time range limited between January 2020 and May 2022. Using a combination of keywords and medical subject headings (MESH), we used vocabulary related to “COVID-19” OR “SARS-CoV-2” AND “NASH” OR “NAFLD” OR “Non-alcoholic Fatty Liver Disease” OR “Fatty Liver” OR “Metabolic Syndrome” in our search (Appendix A).

Six authors (G.J., R.Q., K.R., A.F., A.B., and S.R.) were involved in the inclusion of the studies. After removing duplicates using Endnote, two authors completed the title and abstract screening independently using Rayyan software (https://rayyan.ai/). Studies satisfying the inclusion criteria were retrieved and screened for full-text eligibility. Conflicts between either of the two authors in the inclusion phase were resolved through discussion or later by an additional third arbiter in the case consensus could not be reached.

We included retrospective and prospective observational studies that studied ICU outcomes. We combined NAFLD/MAFLD patients as one group since recent studies are emerging that non-alcoholic fatty liver disease (NAFLD) is related to metabolic syndrome, both mutually and bi-directionally. For simplicity, we used MAFLD in the manuscript to describe both NAFLD and MAFLD. We included studies that assessed MAFLD using lab assessments (FIB-4, APRI, FIBROSIS score, HSI index, etc.), non-invasive imaging (Elastography, Liver Ultrasound or CT scan, MR elastography, and liver stiffness measurement), or liver biopsy. For studies that diagnosed MAFLD using laboratory values, we only included studies that measured MAFLD before index admission since acute COVID infection can elevate lab values, leading to incorrect diagnosis as MAFLD.

We excluded (a) survey studies, (b) case reports, (c) case series, (d) literature reviews, (e) systematic reviews, (f) meta-analyses, (g) single-arm studies, (h) studies that do not measure MAFLD/fatty liver, (i) not a retrospective or prospective study, (j) randomized controlled trials, (k) studies not related to COVID-19 patients, (l) not in the English language, (m) animal studies, (n) abstract only studies, and (o) conference articles. We also excluded studies that did not mention the method of MAFLD assessment.

We also searched systematic reviews and meta-analyses obtained from our search criteria for eligible articles and included eligible studies based on our study criteria. We included studies that compared no or mild MAFLD patient outcomes to moderate or severe MAFLD patient outcomes.

The severity of COVID-19 was defined according to the guidelines on the diagnosis and treatment of COVID-19 issued by the National Health Commission of China [20] for studies originating from China. For the other studies, the severity criteria mentioned in the manuscript were reviewed, and those with similar criteria were included. Severe cases were defined as (1) respiratory distress (≥30 breaths/min), (2) pulse oxygen saturation ≤ 93%, (3) arterial partial pressure of oxygen (PaO_2_)/fraction of inspired oxygen (FiO_2_) < 300 mmHg, and (4) death, hospitalization, oxygen requirement, intensive care unit [ICU] admission, the requirement of vasopressors, or mechanical ventilation. When there were multiple publications with overlapping study populations, we included the one with the greater sample size. We grouped studies for the meta-analysis based on the ICU admission rate, mechanical ventilation, and severity of disease.

We calculated the adjusted odds ratios of the studies to remove the confounding effect of variables that might influence the association between MAFLD and ICU outcomes. Factors like age and obesity, and comorbidities like diabetes and hypertension, have also been reported to have a link with worse ICU outcomes in COVID-19. To establish an independent relationship between MAFLD and COVID-19 disease severity, the need for ICU admission, and mechanical ventilation, we performed an adjusted analysis. The factors adjusted for specific studies and outcomes are provided in Appendix A.

### 2.2. Data Extraction

Three authors (R.Q., A.B., and A.F.) were responsible for data extraction from all the included studies into a pre-piloted data extraction form in Microsoft Excel. A fourth author (G.J.) independently assessed the extracted data for validation. The following was extracted from each study.

General information: author, title, DOI, and year of publication.Study characteristics: study site/country, study period, number of centers, journal, study design, and severe COVID-19 definition.Participant characteristics: number of MAFLD and non-MAFLD patients and the demographic characteristics and method of assessment of MAFLD.Outcomes: ICU admission, invasive mechanical ventilation, and severity of disease in MAFLD and non-MAFLD patients.

### 2.3. Statistical Analysis

Review Manager (RevMan) (computer application) Version 5.4.1 and the Cochrane Collaboration, 2020, were used to assess all results [21]. Using a random-effects model, crude odds ratios (ORs) for each study with corresponding 95% confidence intervals (CIs) were calculated from raw data for events and non-events from each outcome [22]. A *p*-value of <0.05 was considered statistically significant for the analysis. Forest plots were generated to present the results of the meta-analyses. A previously proven technique was used to transform the median to the mean to examine continuous outcomes [23]. The estimates for mean differences were then produced using the random-effects model [22]. To measure study heterogeneity, Cochrane Q and I2 statistics were used [22]. Low-level heterogeneity was defined as I^2^ 20% [22]. The stability of the results was assessed using sensitivity analysis. Egger’s test and funnel plots were used to determine the likelihood of publication bias [24].

### 2.4. Quality Assessment

The NIH scale [25] was used to assess the case-control (Appendix A) and cohort (Appendix A) for appraisal of study quality. We classified studies into three categories: good, fair, or poor. Five authors (G.J., R.Q., A.B., K.R., and H.P.) independently performed the quality assessment of the included studies, and any discrepancies were resolved through discussion.

## 3. Results

### 3.1. Search and Selection

A total of 842 articles were selected for the screening of the title and abstract after deduplication. Ninety-four were chosen for full-text screening, along with eleven studies identified through the back referencing of prior systematic reviews. A total of 28 studies met the inclusion criteria and were included. These studies analyzed ICU admission, invasive mechanical ventilation, and the severity of COVID-19 and were included in the final meta-analysis. These included studies consisted of a total of 42,135 patients, of which 8351 patients were observed to have MAFLD. Figure 1 shows the PRISMA diagram for the included studies.

### 3.2. Characteristics of the Included Studies

There were twenty-eight included studies; five were case-control studies,, one was a cross-sectional study, and twenty-two were cohort studies. Twelve studies were conducted in Asia, nine in North America, and seven in the European region. These studies were conducted between 2020 and 2022. The main characteristics of the included studies are summarized in Table 1.

### 3.3. MAFLD and ICU Admission

Seventeen studies reported the data for ICU admission rates in MAFLD patients, as well as in non-MAFLD controls with COVID-19. Of the total 37,974 patients, 3396 required ICU admission, with 20.41% (1236/6056) of MAFLD patients requiring ICU admission as opposed to 6.77% (2160/31918) of the non-MAFLD patients. The odds ratio was 1.86 for ICU admission, *p* = 0.007, and a 95% confidence interval (95% CI) of [1.18–2.91], with *I*^2^ = 93%. There was a significant increase in the need for ICU admissions in those with MAFLD compared to those without it.

Figure 2 shows the forest plot and meta-analysis of ICU admissions in COVID-19 patients. Figure 3 shows the sensitivity analysis of the included studies. Visual inspection of the standard error plots for the mortality meta-analysis (Appendix A) suggests symmetry without an underrepresentation of studies of any precision. No publication bias was found on the Egger’s test, *p* = 0.354. Seven studies were included in the adjusted analysis to assess the need for ICU admission rates in COVID-19 patients with MAFLD and patients without it. The adjusted odds ratio of all the studies combined was 2.03, *p* < 0.0001, and a 95% confidence interval (95% CI) of [1.47–2.80], with *I*^2^ = 49%. On the sensitivity analysis after excluding the Rentsch 2020 study, the adjusted odds ratio was found to be 1.92, *p* < 0.00001, and a 95% confidence interval (95% CI) of [1.55–2.37], with *I*^2^ = 0%. COVID-19 patients with pre-existing MAFLD had a highly statistically significant need for ICU admission than those without it. Figure 4 shows the adjusted odds ratios for ICU admission in COVID-19 patients with and without MAFLD. Figure 5 shows the adjusted odds ratios forest plot with the sensitivity analysis of the included studies.

### 3.4. MAFLD and Invasive Mechanical Ventilation

Thirteen studies reported the use of invasive mechanical ventilation in MAFLD patients, as well as in non-MAFLD controls with COVID-19. Of the total 37,166 patients, 1676 required mechanical ventilation, with 14.20% (805/5670) of MAFLD patients requiring mechanical ventilation, as opposed to 2.76% (871/31,496) of the non-MAFLD patients. The odds ratio was 2.05 for mechanical ventilation, *p* = 0.02, and a 95% confidence interval (95% CI) of [1.12–3.74], with *I*^2^ = 95%. Figure 6 shows a forest plot and meta-analysis of mechanical ventilation in COVID-19 patients. There was a significant increase in the need for invasive mechanical ventilation in COVID-19 patients with MAFLD, as opposed to those without it. Figure 7 shows the sensitivity analysis of the included studies. Visual inspection of the standard error plots for the mortality meta-analysis (Appendix A) suggests symmetry without an underrepresentation of studies of any precision. No publication bias was found on the Egger’s test, *p* = 0.071. Six studies were included in the adjusted analysis to assess the need for mechanical ventilation in COVID-19 patients with MAFLD and patients without it. The adjusted odds ratio of all the studies combined was 2.06, *p* < 0.0001, and a 95% confidence interval (95% CI) of [1.45–2.92], with *I*^2^ = 46%. On the sensitivity analysis after excluding the Trivedi 2021 study, the adjusted odds ratio was found to be 2.21, *p* < 0.00001, and a 95% confidence interval (95% CI) of [1.75–2.79], with *I*^2^ = 0%. COVID-19 patients with pre-existing MAFLD had a greater need for mechanical ventilation than those without it, and it was highly statistically significant. Figure 8 shows the forest plot with adjusted odds ratios for mechanical ventilation in COVID-19. Figure 9 shows the adjusted odds ratios forest plot with the sensitivity analysis of the included studies.

### 3.5. MAFLD and Severity of COVID-19

Fourteen studies reported COVID-19 disease severity in MAFLD patients, as well as in non-MAFLD controls. Of the total 5286 patients, 1623 had severe COVID-19, with 33.17% (901/2716) of MAFLD patients having severe COVID-19 disease, as opposed to 28.09% (722/2570) of the non-MAFLD patients. The odds ratio was 1.59 for disease severity, *p* = 0.010, and a 95% confidence interval (95% CI) of [1.12–2.26], with *I*^2^ = 81%. Figure 10 shows the forest plot and meta-analysis of disease severity in COVID-19 patients with and without MAFLD. There was a significant increase in the severity of COVID-19 disease in those with MAFLD, as opposed to those without it. Figure 11 shows the sensitivity analysis of the included studies. Visual inspection of the standard error plots for the mortality meta-analysis (Appendix A) suggests symmetry without an underrepresentation of studies of any precision. No publication bias was found on the Egger’s test, *p* = 0.000. We also found that Hussain et al. 2021 had comparatively increased events compared to the other studies. The study was conducted on patients with diabetes and NAFLD only. Eleven studies were included to assess disease severity in COVID-19 patients with MAFLD and those without it. The adjusted odds ratio of all the studies combined was 2.78, *p* < 0.00001, and a 95% confidence interval (95% CI) of [2.00–3.88], with *I*^2^ = 81%. On the sensitivity analysis after excluding the Yoo 2021 study, the adjusted odds ratio was found to be 3.00, *p* < 0.00001, and a 95% confidence interval (95% CI) of [2.38, 3.79], with *I*^2^ = 39%. COVID-19 patients with pre-existing MAFLD had more severe disease than those without it, and it was highly statistically significant. Figure 12 shows the forest plot with the adjusted odds ratios for severity in COVID-19 patients. Figure 13 shows the adjusted odds ratios forest plot with the sensitivity analysis of the included studies.

### 3.6. Quality Assessment

The Appendix A contain figures for the quality assessment of our included studies. Three of the case-control studies (Appendix A) were identified to be of good quality and the other two as fair, while no study was found to be of poor quality. None of them was able to recruit a concurrent control or blind the outcome assessors. Only Madan et al. [38] and Tripon et al. [44] discussed reasons for selecting included participants, providing a sample size justification. Trivedi et al.’s study [45] was the only one to include a random selection of participants.

One included a cross-sectional study [36] that was identified as fair quality (Appendix A). For the included cohort studies (Appendix A), 12 of those were rated to be of good quality, while 10 as fair, with no studies rated as poor. Only Kim et al. [37], Yoo et al. [51], and Zhou et al. [53] could provide a sample size justification; additionally, studies scored poorly on the blinding of outcome assessors, with only Marjot et al. [54] and Zhou et al. [53] able to do so. Furthermore, only Huang et al.’s study [35] was able to measure exposure more than once for each person during the study period.

## 4. Discussion

Our systematic review and meta-analysis of twenty-eight studies and 37,974 COVID-19 patients provided a comprehensive assessment of the ICU admission rate, need for invasive mechanical ventilation, and disease severity of COVID-19 in patients with pre-existing fatty liver disease. The current meta-analysis found significantly higher odds of COVID-19-associated ICU admission in patients with pre-existing MAFLD compared to non-MAFLD patients. Analysis of the studies reporting COVID-19 disease severity in pre-existing MAFLD patients found that a significantly higher number of MAFLD patients developed severe infections and had poor prognoses compared to non-MAFLD patients. A similar trend was observed in the need for invasive mechanical ventilation in COVID-19 infection, with a significantly higher number of MAFLD patients requiring ventilatory support compared to non-MAFLD patients. These results align with our previous study evaluating mortality in COVID-19 patients with pre-existing MAFLD. We found no significant difference in those with fatty liver disease than those without it, although MAFLD patients appear to have higher rates of hospital admissions and longer in-hospital stays compared to non-MAFLD patients [55].

Although multiple independent studies, including retrospective cohort studies by Tignanelli et al. [43] and Younossi et al. [52] and prospective observational studies by Mushtaq et al. [40] and Calapod et al. [26], have shown pre-existing MAFLD to influence COVID-19 severity in terms of increased ICU admission rates and/or the need for mechanical ventilation, these studies are limited by their observational nature, pre-existing confounding biases, and heterogeneity in their sample sizes. Our results validated the above findings. SARS-CoV-2 enters the human cells through the angiotensin-converting enzyme-2 (ACE-2) receptors [56]. This entry is facilitated by viral membrane fusion with the infected host cell membrane. This process is assisted by the priming of SARS-CoV-2 spike proteins by a host cell transmembrane protein called the type II transmembrane serine protease (TMPRSS2) [57]. Patients with pre-existing NAFLD are found to have increased expression of ACE 2 receptors, leading to an increased risk of developing severe COVID-19 disease [58]. Additionally, Shao et al. [59] observed a significant increase in the number of TMPRSS2+ cells in cirrhotic livers, which worsened COVID-19 outcomes. The authors also found that pre-existing MAFLD might increase the susceptibility to the SARS-CoV-2 virus due to a higher TMPRSS2+ progenitor cell count.

MAFLD creates a persistent low-grade inflammatory state mediated through insulin resistance and is associated with obesity and diabetes mellitus (DM), which are comorbidities that cause poor outcomes in COVID-19. The chronic inflammatory state alters the immune system’s ability to respond to infection, which could worsen severe COVID-19 infection [16,17]. Pre-existing MAFLD exacerbates the SARS-CoV-2-induced acute inflammatory response to COVID-19 infection in the host body by aggravating proinflammatory cytokine and reactive oxygen species release during active infection [60,61]. Targher et al. [42] studied the relationship between imaging-defined NAFLD and the neutrophil-to-lymphocyte ratio (NLR) in patients with NAFLD. They found that NAFLD patients had an increased NLR and T lymphopenia compared to non-NAFLD patients. Liu et al. [61] studied a cohort of 245 patients with COVID-19 and demonstrated that there was an 8% higher risk of hospital mortality for every unit increase in the NLR. Patients with increased NLRs also had worse hospital outcomes, likely from an increase in the release of proinflammatory cytokines due to the worsening of the inflammatory/cytokine storm during active infection [42,62].

A retrospective study on 202 patients with NAFLD found that patients with NAFLD had a longer period of viral shedding time of 17.5 days compared to patients without NAFLD, who had 12.1 days [16]. Poor containment of the virus within the host body due to defective immune response and systemic inflammation promotes prolonged viral shedding. Furthermore, the suppression of interferon production by the obese microenvironment in metabolic syndrome/NAFLD and the upregulation of the ACE-2 receptor in COVID-19 infection aggravates viral RNA replication. Both, in turn, result in increased infectivity of the virus and severity of the infection [63]. These hypotheses explain the synergistic nature of MAFLD and COVID-19 pathogenesis. Studies have shown evidence that liver diseases and COVID-19 complement each other’s disease course. Existing hepatic steatosis and MAFLD influence COVID-19 disease severity, ICU admission, and invasive mechanical ventilation requirements, while COVID-19 contributes to furthering hepatic injury and worsening the disease course in MAFLD and other liver diseases [64,65]. However, MAFLD often coexists with other entities, like obesity and DM, under the umbrella of metabolic syndrome. The complex interplay between MAFLD and comorbidities like obesity and DM leads to difficulty in establishing a causal link between MAFLD and COVID-19 outcomes exclusively, independent of these comorbidities.

### Strength and Limitations

We followed rigorous methodology and adhered to the PRISMA guidelines. Our study was registered in PROSPERO. Since most of the included studies were retrospective case-control and cohort studies, this could lead to a risk of bias, particularly in the absence of adjusting for confounders. MAFLD patients have additional underlying medical conditions or co-morbidities that are components of metabolic syndrome. These can interfere with the outcomes we studied. The retrospective nature of most of the included studies does not imply a causal relationship between MAFLD and the studied outcomes. We included studies that measured MAFLD using non-invasive procedures, such as imaging and biomarker/lab diagnosis. Since they are not the gold standard for the diagnosis of MAFLD, there may be misclassification of patients. The definitions of MAFLD were similar, but the diagnostic methods varied across different studies, so there may be some misdiagnoses that can influence the results of the study. However, we included studies that only compared absent or mild MAFLD to moderate or severe MAFLD to maintain homogeneity. We excluded studies on MAFLD patients who also had other causes for liver diseases, including alcoholic liver disease. We considered MAFLD patients in the same group; however, there may be important differences between these two groups that may alter the outcomes. Our meta-analysis observed the per-study protocol. Significant differences were observed for the various outcomes measured in our study population. However, the existence of statistical heterogeneity in our results should be taken into consideration. We also excluded studies in a language other than English due to difficulty with translation and interpretation of results. This can introduce bias in our study results. The studies included in our meta-analysis belonged to different countries, with a higher number of studies from China and the USA, but we consider the results to be globally generalizable. Our study aims for hypothesis generation and to encourage further research in MAFLD patients to better understand the disease pathophysiology and patient risk profiles.

## 5. Conclusions

Our meta-analysis suggests that MAFLD patients develop more severe COVID-19 infection with higher rates of ICU admissions, severity, and greater requirement of invasive mechanical ventilation than non-MAFLD-infected patients. Fatty liver may be associated with worse clinical outcomes in COVID-19.

## Figures and Tables

**Figure 1 medicina-59-01239-f001:**
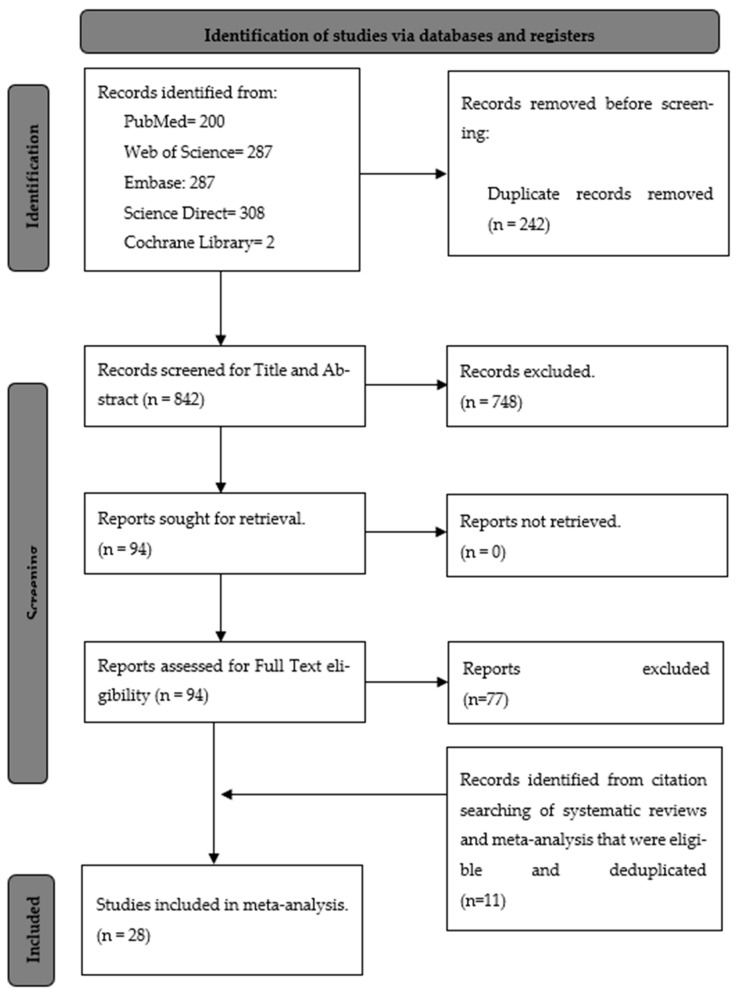
PRISMA flowchart outlining the study search.

**Figure 2 medicina-59-01239-f002:**
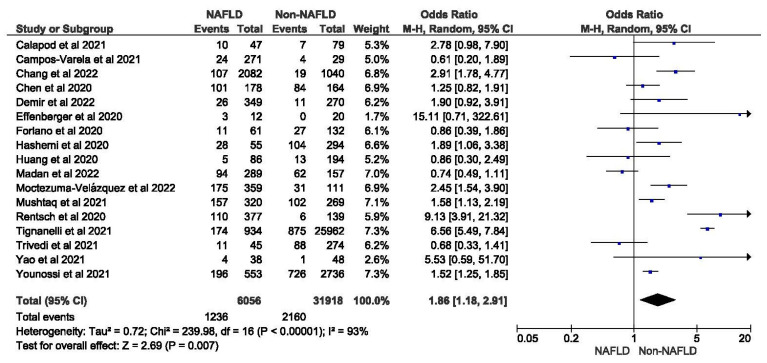
Forest plot for ICU admission in COVID-19 patients with and without MAFLD [26,27,28,29,30,31,32,34,35,38,40,41,43,45,47,50,52].

**Figure 3 medicina-59-01239-f003:**
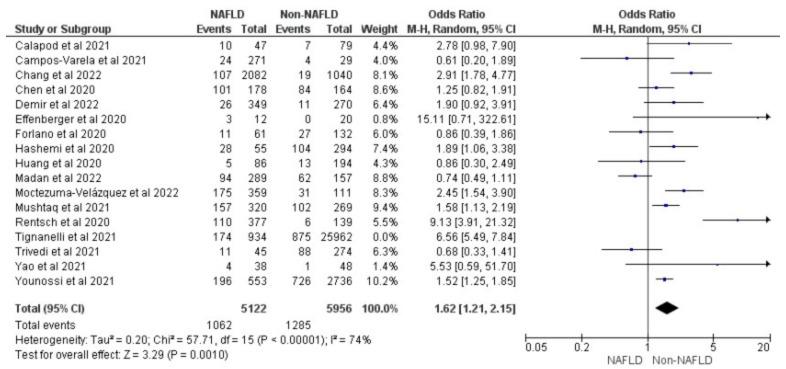
Forest plot for ICU admissions in COVID-19 patients with and without MAFLD sensitivity analysis after excluding Tignanelli et al., 2021 [26,27,28,29,30,31,32,34,35,38,40,41,43,45,47,50,52].

**Figure 4 medicina-59-01239-f004:**
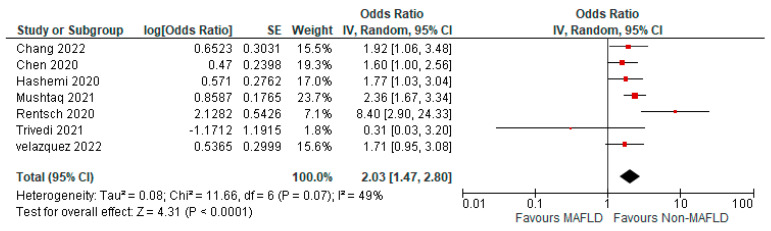
Forest plot with adjusted odds ratios for ICU admission in COVID-19 patients with and without MAFLD [28,29,34,40,41,45,47].

**Figure 5 medicina-59-01239-f005:**
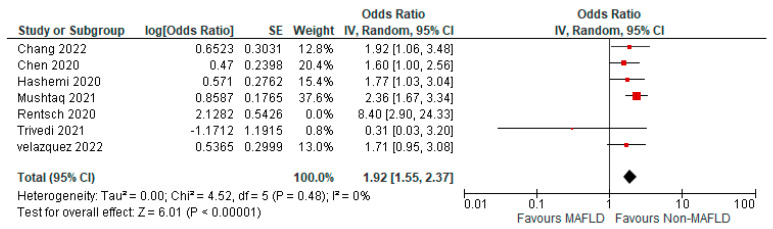
Forest plot with adjusted odds ratios for ICU admission in COVID-19 patients with and without MAFLD sensitivity analysis after excluding Rentsch et al. [28,29,34,40,41,45,47].

**Figure 6 medicina-59-01239-f006:**
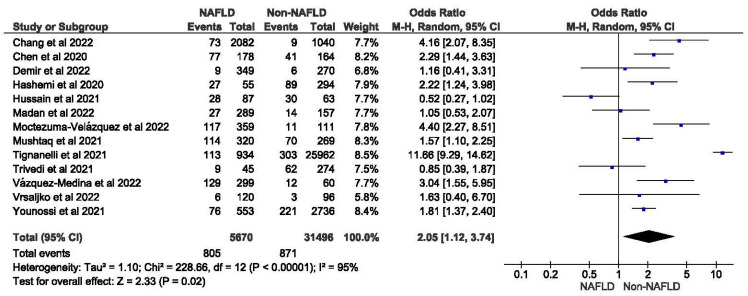
Forest plot and meta-analysis of the need for invasive mechanical ventilation in COVID-19 patients with and without MAFLD [28,29,30,34,36,38,40,43,45,46,47,48,52].

**Figure 7 medicina-59-01239-f007:**
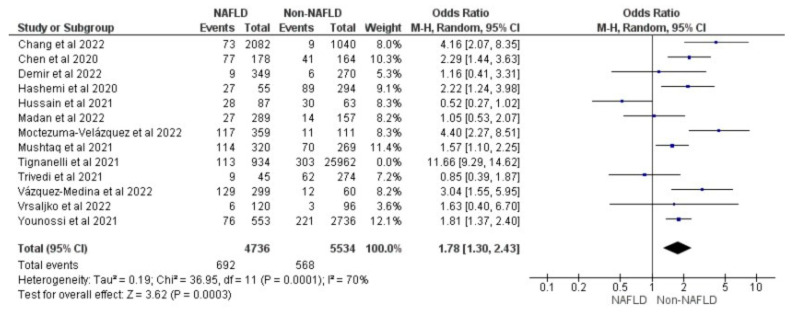
Forest plot and meta-analysis of the need for invasive mechanical ventilation in COVI-19 patients with and without MAFLD sensitivity analysis after excluding Tignanelli et al. [28,29,30,34,36,38,40,43,45,46,47,48,52].

**Figure 8 medicina-59-01239-f008:**
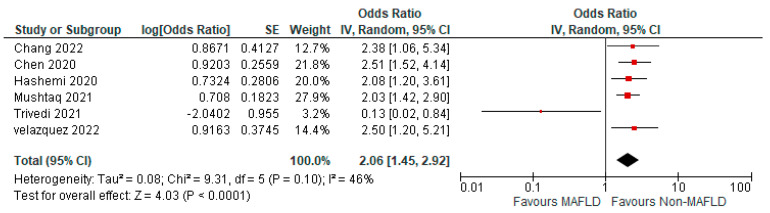
Forest plot with adjusted odds ratios for mechanical ventilation in COVID−19 patients with and without MAFLD [28,29,34,40,45,47].

**Figure 9 medicina-59-01239-f009:**
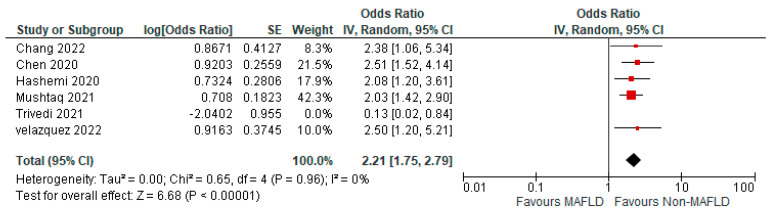
Forest plot with adjusted odds ratios for mechanical ventilation in COVID−19 patients with and without MAFLD sensitivity analysis after excluding Trivedi et al. [28,29,34,40,45,47].

**Figure 10 medicina-59-01239-f010:**
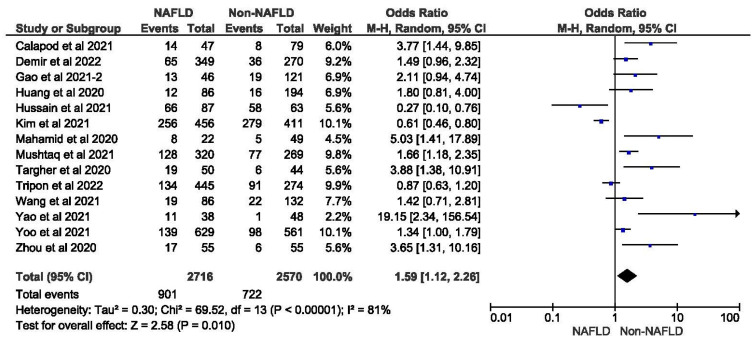
Forest plot and meta-analysis of disease severity in COVID-19 patients with and without MAFLD [26,30,33,35,36,37,39,40,42,44,49,50,51,53].

**Figure 11 medicina-59-01239-f011:**
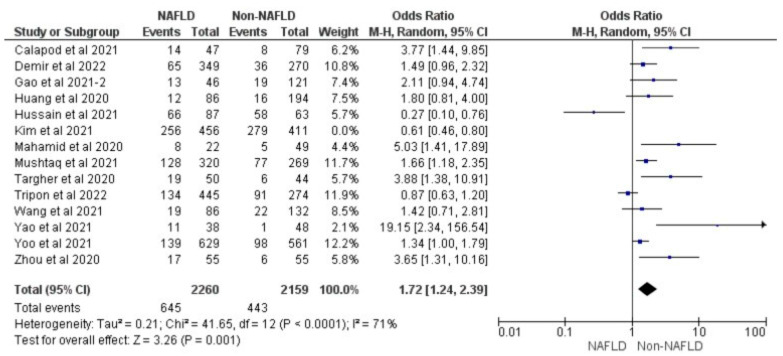
Forest plot and meta-analysis of disease severity in COVID-19 patients with and without MAFLD sensitivity analysis after excluding Kim et al. [26,30,33,35,36,37,39,40,42,44,49,50,51,53].

**Figure 12 medicina-59-01239-f012:**
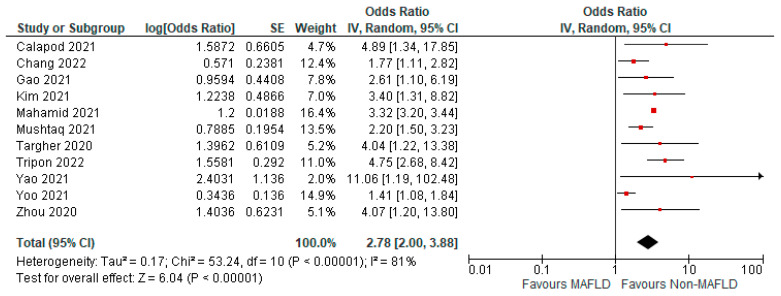
Forest plot with adjusted odds ratios for severity in COVID-19 patients with and without MAFLD [26,28,33,37,39,40,42,44,50,51,53].

**Figure 13 medicina-59-01239-f013:**
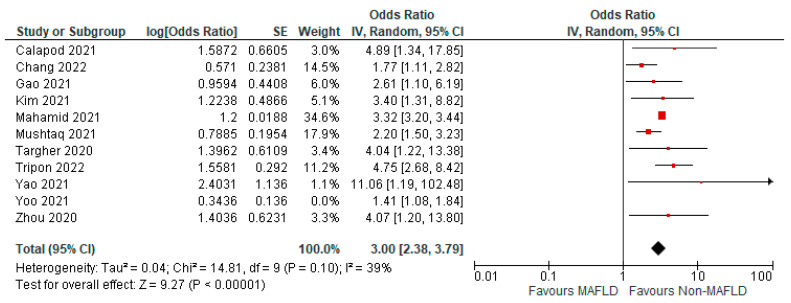
Forest plot with adjusted odds ratios for severity in COVID-19 patients with and without MAFLD sensitivity analysis after excluding Yoo et al. [26,28,33,37,39,40,42,44,50,51,53].

**Table 1 medicina-59-01239-t001:** Main characteristics of the included studies in the systematic review and meta-analyses.

Serial No.	First Author	Country	Year of Publication	Study Design	MAFLD No.	Non-MAFLD No.	Mean Age	±SD	Male (%)	Measure of FIB-4	Measure of FIB-4
1	Calapod [26]	Romania	2021	Prospective descriptive study	47	79	66.32	13.72	57.20%	Imaging evidence (ultrasound or computer tomography)	Biochemical enzymes (LFT) within the past 12 months
2	Campos [27]	Spain	2021	Prospective observational (cohort) study	271	29	55.25	11.69	49%	Liver steatosis by hepatic steatosis index (HSI)	Transient elastography (TE) by controlled attenuation parameter (CAP)
3	Chang [28]	South Korea	2022	Retrospective cohort study	2082	1040	-	-	30.72%	FLI index	
4	Chen [29]	USA	2020	Retrospective single-center cohort study	178	164	62.6	15.6	53.50%	Liver steatosis	Imaging evidence of steatosis > 30 days before COVID-19 diagnosis, or hepatic steatosis index (HSI)
5	Demir [30]	Turkey	2022	Retrospective cohort study	349	270	51.6	9.65	58.60%	FIB-4 index	
6	Effenberger [31]	Austria	2020	Prospective study	12	20	-	-	40.62%	Liver stiffness measurements and CAP with a fibro scan	Liver and spleen sonography and elastography
7	Forlano [32]	UK	2020	Retrospective cohort study	61	132	-	-	60%	Fibrosis-4 index (FIB-4)	Imaging (either ultrasound or computerized tomography) or past medical history
8	Gao [33]	China	2021	Cohort	46	121	49		42.50%	Hepatic steatosis on CT scan	
9	Hashemi [34]	USA	2020	Retrospective cohort	55	294	63.4	16.5	55.4%	Hepatic steatosis on any prior imaging studies or liver histology	
10	Huang [35]	China	2020	Retrospective cohort study	86	194	43.6	17.8	52.10%	Hepatic steatosis index (HSI)	
11	Hussain [36]	Pakistan	2021	Cross-sectional study	87	63	59.73	11.35	56%	Clinical parameters, like hepatomegaly, and lab parameters, like AST, ALT	
12	Kim [37]	USA	2021	Observational cohort study	456	411	56.9	14.5	54.70%	Fibrosis by magnetic resonance elastography	Fibro scan, fibrosis-4, or biopsy
13	Madan [38]	India	2022	Case-control study	289	157	-	-	64.5%	Liver attenuation index (LAI)	
14	Mahamid [39]	Israel	2021	Retrospective, case-control study	22	49	51	21.7	28.20%	CT scan	Medical records documentation
15	Mushtaq [40]	Qatar	2021	Prospective study	320	269	-	-	84.71%	Hepatic steatosis index (HSI)	
16	Rentsch [41]	USA	2020	Retrospective cohort study	377	139	65.8	7.8	95.4	FIB-4 index	
17	Targher [42]	China	2020	Cohort study	50	44	-	-	48%	Fibrosis-4 (FIB-4)	NAFLD fibrosis score (NFS)
18	Tignanelli [43]	USA	2021	Retrospective cohort study	934	25,962	51	23.7	56%	Elevated Alanine Aminotransferase (ALT) Level On 3 Separate Dates	
19	Tripon [44]	France	2022	Retrospective study	445	274	61.35	12.45	58.80%	Hepatic steatosis index	NAFLD fibrosis score (NFS)
20	Trivedi [45]	USA	2021	Retrospective Case-Control study	45	274	65 (median)	-	50%	Abdominal imaging (computed tomography (CT), magnetic resonance imaging (MRI), or ultrasound)	
21	Vazquez-Medina [46]	Mexico	2022	Retrospective case control study	299	60	54.3	14.69	22.01%	FIB-4 index	
22	Moctezuma-Velazquez [47]	Mexico	2022	Retrospective cohort study	359	111	51.6	14.8	63%	CT scans	
23	Vrsaljko [48]	Republic of Croatia	2022	Prospective observational (cohort) study	120	96	59.3	12.6	63.43%	Ultrasound	Difference between liver and spleen computed tomography (CT) attenuation
24	Wang [49]	China	2021	Retrospective cohort study	86	132	-	-	50.40%	Ultrasound parameters	
25	Yao [50]	China	2021	Retrospective Cohort study	38	48	43.2	15.45	58.10%	Hepatic steatosis index (HSI)	NAFLD fibrosis score (NFS)
26	Yoo [51]	South Korea	2021	Retrospective Cohort study	629	561	-	-	-	HSI, FLI, claims based on NAFLD	
27	Younossi [52]	USA	2021	Observational cohort study	553	2736	-	-	49.55%	Abdominal imaging, magnetic resonance imaging, computer tomography, ultrasound	
28	Zhou [53]	China	2020	Cohort study	55	55	42.1	11.4	74.50%	Computed tomography (CT)	

## Data Availability

The data presented in this study are available within the article and its Appendix A.

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
