# Peer review of "Association of Non-Alcoholic Fatty Liver Disease and Metabolic-Associated Fatty Liver Disease with COVID-19-Related Intensive Care Unit Outcomes: A Systematic Review and Meta-Analysis"

_medicina, 2023, doi:10.3390/medicina59071239_

Round 1
Reviewer 1 Report
The present study is an updated meta-analysis on the effects of MAFLD on Covid outcomes. The topic is not really new, however it is interesting and reports, again, an association of MAFLD with worse Covid outcomes. My main comments regarding this study:
- For practical aspects I would unify terminology regarding MAFLD and NAFLD. Since there’s no clear consensus regarding this entity (known as NASH not so long ago), I think there’s no problem if referring to it as either MAFLD or NAFLD. It will greatly improve readability and I think no one is going to be surprised.
- Figure 1 is incomplete since lacking the 11 studies added from references in previous MA and giving the final number of studies.
- Vasopressor utilization is only reported by one study, so there’s no place for this variable.
- Mechanical ventilation is referring to invasive MV, non-invasive MV, or both? Please clarify and provide a differential analysis on invasive and non-invasive MV If available.
- Severity of Covid: this is a very ill-defined variable and I am not clear it since probably mimicking or resulting from a combination of previous variables (mechanical ventilation, vasoactive drugs, ICU admission, etc). In fact, any patients requiring ICU admission with a PaFiO2 < 200 could be probably considered as severe. This variable must be clearly defined and accordingly reported. It must be presented in Methods and Results section. It makes no sense presenting severity data while not defining it.
- Mortality must be presented. This is the hardest and most objective endpoint to be provided in this MA.
- Are data regarding grade of obesity available in the selected studies? This could be an interesting point to see if it modifies the MAFLD effects on outcomes…
- I have had no time to review all of the included studies in the MA. Could authors confirm that association of MAFLD with each association was adjusted for other covariables/confounders (as reported in the original study)? If there are non-adjusted studies, could authors provide sub-analysis including only the “adjusted” studies?
Author Response
The present study is an updated meta-analysis on the effects of MAFLD on Covid outcomes. The topic is not really new; however it is interesting and reports, again, an association of MAFLD with worse Covid outcomes. My main comments regarding this study:
- For practical aspects I would unify terminology regarding MAFLD and NAFLD. Since there’s no clear consensus regarding this entity (known as NASH not so long ago), I think there’s no problem if referring to it as either MAFLD or NAFLD. It will greatly improve readability and I think no one is going to be surprised.
Thank you for the valuable feedback.
We have changed NAFLD or MAFLD to MAFLD in the entire manuscript after the methods. We also mentioned in the methods “For simplicity, we used MAFLD in the manuscript to describe NAFLD and MAFLD.”
- Figure 1 is incomplete since lacking the 11 studies added from references in previous MA and giving the final number of studies.
Thank you. We edited the figure now with a total of 28 studies.
- Vasopressor utilization is only reported by one study, so there’s no place for this variable.
Thank you for the comment. We agree. We removed the variable.
- Mechanical ventilation is referring to invasive MV, non-invasive MV, or both? Please clarify and provide a differential analysis on invasive and non-invasive MV If available.
Mechanical Ventilation only includes invasive mechanical ventilation. We thank the authors for their insight. We have mentioned this in our manuscript.
- Severity of Covid: this is a very ill-defined variable and I am not clear it since probably mimicking or resulting from a combination of previous variables (mechanical ventilation, vasoactive drugs, ICU admission, etc). In fact, any patients requiring ICU admission with a PaFiO2 < 200 could be probably considered as severe. This variable must be clearly defined and accordingly reported. It must be presented in Methods and Results section. It makes no sense presenting severity data while not defining it.
We thank the reviewer for the feedback and agree with the reviewer's comments. We have revised the last paragraph of the search and selection in the methods section “The severity of COVID-19 was defined according to the guidelines on the Diagnosis and Treatment of COVID-19 issued by the National Health Commission of China[20] for studies originating from China. For the other studies, the severity criteria mentioned in the manuscript were reviewed, and those with similar criteria were included. Severe cases were defined as: 1. respiratory distress (>=30 breaths/min) 2. pulse oxygen saturation ≤ 93% 3. arterial partial pressure of oxygen (PaO2)/ fraction of inspired oxygen (FiO2) <300 mmHg 4. death, hospitalization, oxygen requirement, intensive care unit [ICU] admission, the requirement of vasopressors, or mechanical ventilation.
- Mortality must be presented. This is the hardest and most objective endpoint to be provided in this MA.
We have mortality results presented in the first manuscript we published in the World Journal of Gastroenterology. We found that NAFLD is not associated with mortality in COVID-19 patients. ( The odds ratio (OR) was 1.38 for mortality with a 95% confidence interval (95%CI) = 0.97-1.95 and P = 0.07)
MAFLD patients appear to have higher rates of hospital admissions and longer in-hospital stays without any increase in mortality compared to non-NAFLD/MAFLD patients.
Jagirdhar GSK, Qasba RK, Pattnaik H, Rama K, Banga A, Reddy ST, Flumignan Bucharles AC, Kashyap R, Elmati PR, Bansal V, Bains Y, DaCosta T, Surani S. Association of non-alcoholic fatty liver and metabolic-associated fatty liver with COVID-19 outcomes: A systematic review and meta-analysis. World J Gastroenterol 2023; 29(21): 3362-3378 [DOI: 10.3748/wjg.v29.i21.3362]
We thank the author for mentioning this. We have mentioned these results in the current manuscript in the discussion.
- Are data regarding grade of obesity available in the selected studies? This could be an interesting point to see if it modifies the MAFLD effects on outcomes…
No, unfortunately, they are not available, but that would be an interesting measure.
- I have had no time to review all of the included studies in the MA. Could authors confirm that association of MAFLD with each association was adjusted for other covariables/confounders (as reported in the original study)? If there are non-adjusted studies, could authors provide sub-analysis including only the “adjusted” studies?
Thank you, reviewer, for the important feedback. We presented the unadjusted outcomes in the manuscript. After the reviewer's comments, we have now included adjusted outcomes in the manuscript and factors adjusted for in the supplementary file.
Reviewer 2 Report
Dear colleagues!
I read with interest your manuscript "Association of Non-Alcoholic Fatty Liver Disease and Metabolic-Associated Fatty Liver Disease with COVID-19 Related Intensive Care Unit Outcomes: A Systematic Review and Meta-Analysis". The paper is based on thoroughly performed literature review with strict rules for paper selection. In total, it included the data of 28 studies with a population of 37 974 COVID-19 patients. Methodological approaches used in the study are appropriate and allow to get the answer to the main question on the association of pre-existing NAFLD/MAFLD with ICU admission, mechanical ventilation, and severity of COVID-19. The main findings were that the presence of NAFLD/MAFLD is associated with ca. 1.7 greater odds of admission to intensive care unit, severe manifestations of COVID-19 and mechanical ventilation. In my opinion, all strengths and limitations of the study are correctly mentioned in the discussion. I found no major flaws.
The minor flaws that don't affect the scientific quality of the paper are related to imaging quality (partially hidden text in the boxes of the figure 1, font size on the rest of figures).
According to the figures 6&7 the data by Hussain [38] are somewhat in conflict with most of the other studies. Can this be mentioned/explained in the discussion?
Author Response
I read with interest your manuscript "Association of Non-Alcoholic Fatty Liver Disease and Metabolic-Associated Fatty Liver Disease with COVID-19 Related Intensive Care Unit Outcomes: A Systematic Review and Meta-Analysis". The paper is based on thoroughly performed literature review with strict rules for paper selection. In total, it included the data of 28 studies with a population of 37 974 COVID-19 patients. Methodological approaches used in the study are appropriate and allow us to get the answer to the main question on the association of pre-existing NAFLD/MAFLD with ICU admission, mechanical ventilation, and severity of COVID-19. The main findings were that the presence of NAFLD/MAFLD is associated with ca. 1.7 greater odds of admission to intensive care unit, severe manifestations of COVID-19 and mechanical ventilation. In my opinion, all strengths and limitations of the study are correctly mentioned in the discussion. I found no major flaws.
The minor flaws that don't affect the scientific quality of the paper are related to imaging quality (partially hidden text in the boxes of the figure 1, font size on the rest of figures).
We thank the reviewer for noticing this. For Figure 1 we have edited to show all the text in the boxes. For the rest of the figures, we have provided clear images and increased the font size.
According to figures 6&7 the data by Hussain [38] are somewhat in conflict with most of the other studies. Can this be mentioned/explained in the discussion?
We thank the author for this observation. The authors have provided only results for patients with Diabetes and NAFLD. This is the reason for increased event outcomes. We have mentioned this in the severity outcomes in the results section now.
Reviewer 3 Report
I found this systematic research and the meta analysis very well described. The way it was perfomed was very methodic and properly design, and also possible pitfalls and limitations are highlighted which is very much appreciated.
The outcome of this study is not highly novel since, as authors already mentioned in their manuscript, there were many previous evidences between NAFLD with a worse prognosis of COVID-19 disease. However, since they focus on some intensive care unit outcomes which are very specific, this gives some novelty to the study, and the conclusions obtained seem robust given the amount of patients included in their analysis.
I found some minor spelling mistakes that need to be corrected:
-In affiliation 4, the name of the country is missing (India).
-On line 63 the word "with" should be removed.
-On table 1, on reference 9, there is a "%" missing on 55.4 column.
-On Figure 7, the capital letter of "Sensitivity" should be removed.
I would also recommend the authors to include in the keywords, some aspects related to the real goal of their research, which is the study of the intensive care unit outcomes .
Author Response
I found this systematic research and the meta-analysis very well described. The way it was performed was very methodic and properly design, and also possible pitfalls and limitations are highlighted which is very much appreciated.
The outcome of this study is not highly novel since, as authors already mentioned in their manuscript, there were many previous evidence between NAFLD with a worse prognosis of COVID-19 disease. However, since they focus on some intensive care unit outcomes which are very specific, this gives some novelty to the study, and the conclusions obtained seem robust given the amount of patients included in their analysis.
I found some minor spelling mistakes that need to be corrected:
-In affiliation 4, the name of the country is missing (India).
We thank the reviewer for the comment. We have mentioned Green Life Medical College and Hospital, Dhaka, 1205, Bangladesh for affiliation 4. We hope this is correct. Please correct us if not.
-On line 63 the word "with" should be removed.
We have removed it. Thank You
-On table 1, on reference 9, there is a "%" missing on 55.4 column.
We thank the author for this. We have added the %
-On Figure 7, the capital letter of "Sensitivity" should be removed.
Agree. Thank you. We have edited this.
I would also recommend the authors to include in the keywords, some aspects related to the real goal of their research, which is the study of the intensive care unit outcomes .
Agree. We have added Critical care; intensive care units; in the keywords. Thank You